# Metagenomic Analysis of a Continuous-Flow Aerobic Granulation System for Wastewater Treatment

**DOI:** 10.3390/microorganisms11092328

**Published:** 2023-09-15

**Authors:** Alison T. Gomeiz, Yewei Sun, Aaron Newborn, Zhi-Wu Wang, Bob Angelotti, Benoit Van Aken

**Affiliations:** 1School of Systems Biology, George Mason University, 10900 University Blvd, Manassas, VA 20110, USA; agomeiz@gmu.edu; 2Hazen and Sawyer, 4035 Ridge Top Road, Fairfax, VA 22030, USA; ysun@hazenandsawyer.com; 3Department of Chemistry and Biochemistry, George Mason University, 4400 University Dr, Fairfax, VA 22030, USA; anewborn@gmu.edu; 4Department of Biological Systems Engineering, Virginia Tech, 1230 Washington St. SW, Blacksburg, VA 24061, USA; wzw@vt.edu; 5Upper Occoquan Service Authority, 14631 Compton Rd, Centreville, VA 20121, USA; bob.angelotti@uosa.org

**Keywords:** aerobic granulation, plug flow reactor—PFR, metagenomics, extracellular polymeric substances—EPS, filamentous bacteria

## Abstract

Aerobic granulation is an emerging process in wastewater treatment that has the potential to accelerate sedimentation of the microbial biomass during secondary treatment. Aerobic granulation has been difficult to achieve in the continuous flow reactors (CFRs) used in modern wastewater treatment plants. Recent research has demonstrated that the alternation of nutrient-abundant (feast) and nutrient-limiting (famine) conditions is able to promote aerobic granulation in a CFR. In this study, we conducted a metagenomic analysis with the objective of characterizing the bacterial composition of the granular biomass developed in three simulated plug flow reactors (PFRs) with different feast-to-famine ratios. Phylogenetic analyses revealed a clear distinction between the bacterial composition of aerobic granules in the pilot simulated PFRs as compared with conventional activated sludge. Larger and denser granules, showing improved sedimentation properties, were observed in the PFR with the longest famine time and were characterized by a greater proportion of bacteria producing abundant extracellular polymeric substances (EPS). Functional metagenomic analysis based on KEGG pathways indicated that the large and dense aerobic granules in the PFR with the longest famine time showed increased functionalities related to secretion systems and quorum sensing, which are characteristics of bacteria in biofilms and aerobic granules. This study contributes to a further understanding of the relationship between aerobic granule morphology and the bacterial composition of the granular biomass.

## 1. Introduction

The secondary (or biological) treatment of contemporary wastewater treatment plants (WWTPs) accounts for about 90% of organic material removal. In conventional WWTPs, the secondary treatment utilizes activated sludge (AS) technology, which is based on the formation of small microbial aggregates, termed ‘flocs’, ranging roughly from 1 to 100 μm in diameter [1]. The loose structure and poor sedimentation characteristics of the AS render difficult the efficient separation of the biomass from the treated water, which is essential to producing good-quality effluent and recycling the biomass. Therefore, a major disadvantage of the AS system is the requirement for large and energy-consuming secondary clarifiers to achieve gravity separation. To address this issue, membrane-based bioreactors have been tested, but they are not widely used in full-scale operations because of difficulties involving high capital and energy costs, and fouling problems. Recently, reactor design and operations have shown the possibility of producing a microbial biomass in the form of aerobic granular sludge, which possesses good settling properties and constitutes a promising technology for wastewater treatment [2]. Bacterial flocs need to be separated from the treated water in large settlers. Aerobic granulation provides a promising alternative to traditional AS systems because it results in the formation of larger and denser bacterial aggregates termed ‘granules’. Granules range roughly from 300 μm to 5 mm, therefore increasing biomass retention and allowing for smaller sedimentation volumes and shorter settling times [2,3]. Other advantages of aerobic granulation include increased metabolic functions, a compact biomass structure, and protection of the microbial community against unfavorable conditions, such as toxicity shocks [4,5,6]. Aerobic granular sludge differs from traditional AS in its denser, more spherical morphology. The formation of aerobic granules is believed to be associated with high levels of extracellular polymeric substances (EPS) produced by bacteria under appropriate conditions. Unfortunately, aerobic granulation has been shown to be difficult to achieve in the continuous flow reactor (CFR) systems that are overwhelmingly used in most modern WWTPs. Despite over twenty years of research since its discovery, aerobic granulation can only take place efficiently in sequential batch reactors (SBRs), which are generally incompatible with large-scale WWTPs [7]. Therefore, there is high interest in developing a continuous flow system capable of facilitating aerobic granulation of microorganisms in secondary wastewater treatment systems.

A recent literature review on aerobic granulation reveals that the alternation of phases of nutrient abundance and depravation in wastewater is necessary for successful granulation in SBRs [8]. It is thought that feasting periods (i.e., abundant nutrient conditions) followed by adequate periods of famine (i.e., low nutrient availability) provide the appropriate conditions for microbial communities to form aerobic granules. Several studies have determined that successful aerobic granulation in SBRs requires the biomass retention time under feast conditions to be relatively small as compared with the retention time under famine conditions (ratio feast to famine retention time <0.5) [9,10,11,12,13].

The physical and biochemical benefits of aerobic granulation are attributable to the biofilm-like phenotype of the aggregates, which are encapsulated in EPS [5,7,13]. Biofilm formation is a complex process that is regulated by bacterial signal transduction pathways designed to detect environmental stressors like nutrient deprivation [13,14]. As such, key groups of bacteria capable of synthesizing EPS for biofilm formation have been observed in higher abundance in SBR systems with successful aerobic granulation [15,16,17,18,19,20,21,22]. In a pilot study, we recently demonstrated that aerobic granulation can be implemented in a simulated plug flow reactor (PFR) using real domestic wastewater [23,24]. It was suggested that successful granulation was achieved through the induction of sufficiently low feast/famine conditions (ratio feast-to-famine retention time = 0.33). The pilot systems that were used are simulated PFRs consisting of a suite of continuously stirred tank reactors (CSTRs) connected in series such that nutrient availability decreased from one reactor to the next, mimicking the concentration gradient observed along a real PFR. Currently, aerobic granulation cannot be efficiently achieved in conventional CSTRs. However, it has been shown that aerobic granulation can be induced by the alternation of feast and famine conditions in a PFR [24]. In our study, we used a suite of CSTRs in series to simulate a PFR system. The feast/famine conditions in the PFR system were modulated by varying the number of CSTRs (i.e., chambers) from 4 to 8. CSTRs in series have been largely used to simulate PFR conditions by dividing the reactor into multiple segments, each with its own conditions [8]. Three different simulated PFRs (4-, 6-, and 8-chamber systems) with decreasing feast-to-famine retention times (1, 0.5, and 0.33, respectively) were used. Morphological analysis of the biomass in the simulated PFRs revealed that feast/famine ratios greater than 0.5 inhibited the formation of aerobic granules [12,24]. However, the response of the microbial community composition of aerobic granules to varying feast/famine conditions was not investigated in depth.

The objective of the present study was to analyze the bacterial community of aerobic granules developed under different feast/famine conditions in simulated PFRs fed with actual domestic wastewater and to subsequently predict the production of EPS through metabolomic profiles. Successful aerobic granulation largely depends on the composition of the bacterial community. In this study, we used next-generation sequencing to establish a relationship between feast/famine conditions, the aerobic granule phenotype, and the predominance of specific bacterial groups. Our results may contribute to a better understanding of the conditions that promote successful aerobic granulation, which is integral to the future implementation of WWTPs.

## 2. Materials and Methods

### 2.1. Simulated PFR Characteristics and Local Domestic Wastewater

Three different simulated PFR systems were utilized in this study, which consisted of several CSTRs connected in series (number of chambers = 4, 6, and 8) (Appendix A). CSTRs connected in series are known to simulate the behavior of a PFR [24]. The systems are described in full detail in Sun et al. (2019). In brief, the working volume of each CSTR was 12.8 L. Each CSTR was continuously aerated from the bottom of the chamber. The systems were fed with primary effluent from the Upper Occoquan Service Authority (UOSA) municipal WWTP located in Centreville, VA, USA. The hydraulic retention time (HRT) in all three simulated PFRs was maintained at 6.5 h, which is similar to the HRT employed in the full-scale AS system at the plant. Aeration was provided in each CSTR chamber at an air flowrate of ~3.2 L min^−1^ to ensure a dissolved oxygen (DO) > 3 mg L^−1^. The flow rate of the systems was 2 L h^−1^, and the average temperature of the primary effluent was 18.3 °C [24]. The average composition of the primary effluent was as follows: pH = 7.3, total COD (tCOD) = 232 mg L^−1^, soluble COD (sCOD) = 86 mg L^−1^, 5-day BOD (BOD_5_) = 116 mg L^−1^, alkalinity = 211 mg L^−1^, total suspended solids (TSS) = 97 mg L^−1^, total Kjeldahl nitrogen (TKN) = 39.0 mg L^−1^, ammonia (NH_3_-N) = 27.7 mg L^−1^, and total phosphorous (TP) = 4.4 mg L^−1^ [24]. The biomass from the simulated PFRs was sampled at steady-state under sterile conditions (after 400 days of operation). Biomass from the regular AS in the same plant was also sampled for comparison [24]. The feast/famine ratio of each PFR was determined using previously described methods [12]. In short, feast conditions were defined by positive or null microbial net growth, while famine conditions were characterized by negative microbial net growth. Three feast/famine ratios were identified based on the number of chambers with high and low nutrient abundance: 1, 0.5, and 0.33 for the 4-, 6-, and 8-chamber PFR, respectively. After operating the PFRs for 4 months, one sample was collected per chamber from each PFR, with triplicate samples in one chamber of the 4- and 6-chamber PFRs (for which a total of 6 and 8 samples were collected, respectively). Two samples were collected from the AS. One sample was collected from a lab-scale nitrifying reactor as an example of a low-diversity microbial community.

### 2.2. 16S rDNA Sequencing

Fifteen (15)-mL samples of mixed liquor were collected from the reactors (the three pilot PFRs and conventional AS) in sterile tubes, and the biomass was concentrated on-site by decantation. Samples were transported to the lab on ice and stored at −80 °C until DNA extraction.

For DNA extraction, biomass samples were thawed on ice and concentrated by centrifugation for 10 min at 10,000 rpm. The pelleted biomass (roughly 250 mg) was extracted using the DNeasy PowerSoil Kit (Qiagen, Germantown, MD, USA), following the manufacturer’s protocol. Homogenization was completed with a BeadBug™ Homogenizer (MilliporeSigma, Burlington, MA, USA) using 30-s pulses at 3000 rpm until no clumps were visible in the sample. DNA samples were assessed for purity and concentration using a NanoDrop™ One*C* (Life Technologies, Carlsbad, CA, USA). Hypervariable regions V3–V4 of the bacterial 16S rDNA gene were amplified using the 341F/785R primer set (5′-CCTACGGGNGGCWGCAG-3′/5′-GACTACHVGGGTATCTAATCC-3′) with Genewiz^®^ partial adapters attached [25]. PCR amplification was conducted on a QuantStudio™ 3 Real-Time PCR System using the Absolute Blue qPCR SYBR Green Mix (Life Technologies, Carlsbad, CA, USA). The PCR conditions were initial denaturation for 30 s at 98 °C, 40 cycles of denaturation for 10 s at 98 °C, annealing/extension for 75 s at 65 °C, and final extension for 5 min at 65 °C. PCR products were purified using the QIAquick PCR Purification Kit (Qiagen, Germantown MD, USA) and quantified using the NanoDrop™ One*C*. Paired-end (2 × 150 bp) sequencing of the 16S rDNA libraries was performed by Genewiz^®^ (South Plainfield, NJ, USA) on an Illumina HiSeq 2000 (Illumina, San Diego, CA, USA).

### 2.3. Metagenomic Data Analysis

Metagenomic analysis was completed with the R package DADA2 [26] and the Python package Cutadapt [27]. Scripts were run on a 16-core EC2 virtual machine accessed from Amazon Web Services. Briefly, primers were trimmed from forward and reverse reads using Cutadapt [27]. Ends of reads were trimmed based on quality score using filterAndTrim, and reads with quality scores <30 (<2% of total reads) were removed from further analysis [26]. The combination of sequence error analysis using learnErrors and dereplicated sequences obtained from the derep function was used in the sample inference step of the DADA function [fix sentence]. The forward and reverse reads were then merged using the mergePairs tool, followed by a chimera removal step employed through the removeBimeraDenovo step [26]. The finalized actual sequence variants (ASVs) were then searched (BLAST) against the SILVA 16S rRNA database (release 138.1) [28].

### 2.4. Diversity and Compositional Abundance Analysis

Differentially abundant taxa were identified using the R package ALDEx2 [29] at each taxonomic level. Alpha diversity was determined through a series of commonly used indices, such as the Shannon diversity index and the Simpson index. Beta diversity analysis was completed using the R package MicrobiotaProcess [30] to generate a principal coordinate analysis (PCoA) plot. The statistical significance of the data was determined using ALDEx2.

### 2.5. Predicted Functional Analysis

The Python script PICRUSt2 was used to predict the functionality of the identified bacterial communities obtained by DADA2 [31]. ASVs were tested against the functional database KEGG [32]. The outputs of PICRUSt2 were displayed in BURRITO, a web server visualization tool used to find relationships between the taxonomic identification and predicted functions of a microbiome [33]. PICRUSt2 outputs were then tested against the KEGG Pathway database to obtain more detailed information on specific proteins related to defined cellular functions. Heatmaps from predicted functional analysis were generated using base R. Statistically significant functions between the 8-chamber PFR and AS datasets were determined using ALDEx2 [33].

## 3. Results and Discussion

In a prior study, we demonstrated that a PFR system simulated by a series of CSTRs promoted aerobic granulation of the biomass [23]. It was shown that the desired morphology of the aerobic granules (i.e., dense, spherical, regular aggregates with high sedimentation velocity) was achieved by the successive establishment of a period of nutrient sufficiency in the first reactor chambers (feast conditions) followed by a longer period of nutrient limitation (famine conditions) in the later chambers. Moreover, extending the famine phase (achieved by increasing the number of reactor chambers) was shown to improve the granule morphology. The best granule morphology (largest granule size and good consistency) was therefore observed in the 8-chamber PFR, which was characterized by a ratio of feast-to-famine retention time of 0.33 [24]. Smaller, less dense flocs were observed in the 4- and 6-chamber PFR systems.

A metagenomic analysis was conducted using biomass from each chamber of the three PFRs and the full-sized AS system at the plant. A total of 25 samples were collected, including 22 from the three PFRs (one sample in each PFR was analyzed in replicates for quality control), 2 from the AS, and 1 from a nitrifying community used as a low-diversity control. The number of reads obtained ranged from 121,300 to 155,100, with mean quality scores ranging from 37.10 to 37.36 and percentage bases over 30 bp ranging from 92.5 to 93.6%.

The Good’s coverage, an alpha diversity estimator used to evaluate the proportion of microbial species in a sample, was higher than 0.99 for all samples, indicating that the sequencing depth was sufficient to achieve good characterization of the bacterial community (Table 1). The Shannon diversity index is commonly used to characterize the diversity of species in a community. It is based on both the richness (number of operational taxonomic units, or OTUs) and evenness (relative abundance of OTUs). The Shannon diversity indexes indicated a higher diversity in the AS samples than in any of the three PFR systems (Table 1), with the 4- and 6-chamber PFRs showing a lower diversity than the 8-chamber PFR. The Shannon diversity index for the AS was on average 6.28 ± 0.03, which was significantly higher (*p* < 0.001) than the index for the 8-chamber PFR: 5.43 ± 0.20, which was itself significantly higher (*p* < 0.001) than the 6- and 4-chamber PFRs (4.52 ± 0.27 and 4.74 ± 0.09, respectively). As a reference, the Shannon diversity index for the nitrifying reactor was much lower and equal to 3.45. All PFRs were initially fed with AS. The reduction of diversity when transitioning from AS to PFR communities was expected and is explained by the selective pressure of the alternation of feast and famine conditions applied to stimulate aerobic granulation. Within the PFRs, the 8-chambered reactor showed the highest diversity, which may reflect the stratification of microbial species inside larger, denser granules as compared with smaller, less compact granules in the 4- and 6-chambered reactors. These trends were also reflected in the Simpson diversity index and evenness, which was expected as these parameters are frequently correlated. Consistent with our results, lab- and pilot-scale studies reported that the formation of aerobic granules was associated with a lower or similar level of bacterial diversity as compared with seed-activated sludge [6]. On the other hand, several reports indicated an increase in bacterial diversity during the aerobic granulation process, which was explained by the longer retention time of the biomass, allowing the development of slow-growing bacteria [34]. Due to their larger size, large aerobic granules may contain different ecological niches at different depths from the surface [6]. This may explain the higher diversity that we observed in our 8-chamber PFR as compared with other systems.

At the phylum level, we observed in all systems a prevalence of *Proteobacteria*, *Bacteroidota*, *Verrucomicrobiota*, and *Campilobacterota* (Figure 1). Although *Proteobacteria* and *Bacteroidetes* are among the most abundant phyla observed in aerobic granules, other typical granule phyla, such as *Actinobacteria* and *Firmicutes*, were among the predominant phyla in our study [6]. *Proteobacteria*, which, in these reactors, involved mostly *Gammaproteobacteria* (69–86% of the *Proteobacteria*), were significantly more abundant in the aerobic granulation PFRs than in the AS. In the aerobic granulation PFRs, the *Gammaproteobacteria* class contained many members of the genus *Sphaerotilus* (18–33% of the *Proteobacteria* in the aerobic granulation PFRs vs. 0.1% in the AS), which consists mainly of filamentous bacteria involved in sludge bulking. Such filamentous bacteria are known to be essential for efficient aerobic granulation by holding bacterial aggregates together [16,35]. *Bacteroidota*, on the other hand, were significantly more abundant in the 8-chamber PFR than in the 4- and 6-chambered AS systems. *Bacteroidota* are associated with high cellular hydrophobicity and the production of EPS, which is necessary for the generation of large, dense granules, as we observed in the 8-chamber PFR [36]. Similarly, *Campilobacterota* (e.g., *Arcobacter* order) include many EPS producers commonly seen in mature aerobic granules [13,37].

Figure 2 shows the abundance of major bacterial orders observed in the three PFRs and the AS. In all systems, we observed a prevalence of *Burkholderiales* (*Gammaproteobacteria*). As indicated above, the majority of *Burkholderiales* in all PFR systems belong to the genus *Sphaerotilus*, which is associated with the production of long filaments (up to 1000 µm). Previous studies have found that these bacteria increase during the transition from AS to aerobic granules, suggesting their involvement in successful aerobic granulation [15]. The formation of aerobic granules is thought to be initiated by the outgrowth of filamentous bacteria, which capture particulate organic matter to be consumed within the bulk granule [12,38,39]. The 8-chamber PFR, with the densest, best-quality granules, contained approximately 18% of *Sphaerotilus* bacteria. Although this genus is commonly associated with sludge bulking in AS [40], it has also been identified in dense microbial aggregates [41]. On the other hand, a higher proportion of the filamentous *Sphaerotilus* was observed in the 4- and 6-chamber PFR than in the 8-chamber PFR. Indeed, the shorter famine phases in the 4- and 6-chamber PFR may result in a higher proportion of filamentous bacteria. However, this higher proportion of *Sphaerotilus* may be too high for optimal granulation, as suggested by morphological analyses showing larger granules in the 8-chamber reactor than in the 4- and 6-chamber reactors [24]. Filamentous bacteria have been suggested to be necessary for aerobic granulation as they constitute a scaffold for granule growth. However, as observed in our study, several authors indicated that excessive filamentous growth results in a loose granule structure, leading granules to collapse [8]. Filamentous granules have also been shown to have poor settling properties, which are undesirable for wastewater treatment operations [8]. We also observed an increase in *Flavobacteriales* as the number of chambers increased, which could be related to the high production of EPS by these organisms [6]. An inverse relationship was detected between the abundance of *Flavobacteriales* and *Sphingobacteriales* in the 8-chamber system and the AS, with *Flavobacteriales* being more abundant in the 8-chamber PFR. A similar shift (i.e., increase of *Flavobacteriales* and decrease of *Sphingobacteriales*) was previously described during the formation of granules and was accompanied by an increase in cell surface hydrophobicity (CSH) and EPS production, which are favorable to granulation [36,42].

*Campylobacterales*, *Sphingomonadales*, and *Pseudomonadales* were also found in much higher abundance in the 8-chamber PFR than in all other systems. The majority of *Campylobacterales* bacteria (74%) belong to the *Arcobacter* genus, which is known to produce large amounts of EPS [6,15,36,43]. Similarly, *Sphingomonadales* and *Pseudomonadales* are also well-known EPS-producing orders [43]. *Thiotrichales*, which were composed exclusively of the *Thiothrix* genus, were more abundant in the PFR systems than in the AS. The *Thiothrix* genus contains filamentous bacteria prevalent in aerobic granules, and it has been linked to EPS production [6,44,45,46]. Multiple studies have reported that aerobic granules contain an abundance of bacteria producing high amounts of EPS, which are involved in the formation of the granule matrix [6]. EPS acts as a cementing material, adhering bacterial cells together. EPS also contains proteins and polysaccharides, which may contribute to the reduction of charge repulsion between cells and favor bridging between aggregates [8].

Figure 3 displays the 16 most abundant genera detected in the 8-chamber PFR as compared with their abundance in the AS control samples. Of these genera, 8 have been previously reported in granular sludge reactors and/or are composed of filamentous organisms and/or producers of EPS: *Flavobacterium*, *Hydrogenophaga*, *Thiothrix*, *Pseudomonas*, *Acidovorax*, *Cloacibacterium*, *Acinetobacter*, and *Aeromonas* [6,16,17,18,19,20,21,22,47,48]. In their review of the bacterial composition of aerobic granules, Xia et al. [6] indicated the prevalence of several of the most abundant genera identified in our study, including *Flavobacterium*, *Thiothrix*, *Acinetobacter*, *Acidovorax*, *Hydrogenophaga*, and *Sphaerotilus*. The significant increases in abundances of these groups, from AS to the 8-chamber PFR, confirm that the low feast/famine ratio strongly selects bacteria favoring granulation, corroborating earlier findings [5,6,24]. Overall, EPS-producing genera account for approximately 50% of all identified genera in the 8-chamber PFR, compared to only 5% in the AS samples. Other genera more abundant in the 8-chamber PFR than in the AS include *Bdellovibrio*, which are motile and flagellated [49], and *Novosphingobium*, which are non-filamentous bacteria known to form microcolonies [50] and have high CSH [51].

Principal coordinate analysis (PCoA) based on bacterial abundances was conducted to identify the pattern of microbiomes developing in the chambers of each SBR system (4, 6, and 8 chambers) and in the AS control (Figure 4). A clear separation between each of the three SBR systems and the AS samples was observed. Prior studies have similarly reported significant changes in the bacterial composition of the aerobic granules and the activated sludge they were derived from [8,52]. The operational taxonomic units 4 (OTU 4) and 5 correlated more strongly with the 8-chamber PFR, which is consistent with the higher abundance of members of the genus *Arcobacter* and the class *Bacteroidia* observed in this system. The *Arcobacter* genus is commonly associated with granulation and has been commonly identified in SBR systems [15,37,53]. The *Bacteroidia* class is characterized by high CSH and enhanced production of EPS associated with successful aerobic granulation [36]. Another influential OTU in the three PFRs, as opposed to AS, is OTU 1, which includes members of the *Sphaerotilus* genus and is frequently detected in aerobic granule samples [6]. The *Hydrogenophaga* genus (OTU 6) and the *Chitinophagales* order (OTU 10), which are EPS producers associated with good-quality granules [6,54], were more strongly correlated with the 4- and 6-chamber PFRs. As expected, the AS control samples distinguished themselves clearly from the three PFR systems. Similar observations are shown on the heatmap presented in Appendix A.

We used the Python script PICRUSt2, which is based on the KEGG pathway database, to predict the functionality of the identified bacterial communities in the 8-chamber PFR and the AS system, respectively (Figure 5) [31]. The trends observed at the taxonomic level can then be used to approximate and quantify the extent of certain biological functions in these microbial communities [32]. Based on the bacterial functionalities, we first observed that the AS controls differentiate clearly from all the compartments of the 8-chamber PFR. No clear pattern of functionalities can be observed from compartments 1 to 8 of the PFR system. This is not surprising as the PFRs are designed as ‘one sludge’ systems, i.e., the biomass travels with the wastewater in the mixed liquor and is only separated from the wastewater after the last chamber to be recycled in the reactor. As compared with AS, the 8-chamber PFR showed higher activity in secretion systems (*ABC transporters* and *transporters*), which are essential components of EPS production and dissemination. In addition, the higher activity in signaling functions (*quorum sensing* and *signal transduction*) in aerobic granules indicates increased cell-to-cell communication, which aligns with the phenotype observed in mature biofilms [14]. Genes responsible for *signal transduction mechanisms* and the *two-component system* were also more abundant in the 8-chamber PFR (although not uniformly in all chambers). These systems are used by bacteria to detect and respond to environmental fluctuations. These responses include biofilm formation, a common response to environmental stressors such as nutrient deprivation. As such, the two-component system has been identified as the main pathway associated with biofilm formation in microbial communities [55]. The increased *flagellar assembly* activity in aerobic granules as compared with AS is also consistent with the role of the flagella in biofilm formation [56]. The higher prevalence of these systems in aerobic granules is then well explained from both a biofilm formation perspective and as a response to famine periods in the 8-chamber PFR system. The taxa that contribute the most substantially to these functions include *Hydrogenophaga*, *Flavobacterium*, *Pseudomonas*, and *Acidovorax*. The KEGG Pathways identified a variety of biofilm formation processes in the 8-chamber PFR, such as the autoinducer-2 (AI-2) quorum sensing pathway that mediates interspecies communication during biofilm formation, acyl-homoserine lactones in gram-negative bacteria such as *Pseudomonas*, and a variety of EPS-producing pathways. Other functions increased in the 8-chamber PFR as compared with the activated sludge include *nitrogen metabolism* and *sulfur metabolism*, particularly denitrification and sulfate reduction. Denitrifiers and sulfate-reducing bacteria are typically found in the anoxic core of aerobic granules. Due to these inner anoxic/anaerobic zones in granules, aerobic granules provide an adequate environment for denitrification and sulfate reduction, which typically do not occur in aerobic AS [15,57]. The taxa that contribute the most substantially to these functions are *Hydrogenophaga* and *Sphaerotilus* (*nitrogen metabolism*) and *Pseudomonas* (*sulfur metabolism*). One important advantage of aerobic granules over conventional AS is the possibility of simultaneous nitrification and denitrification, which may result in complete ammonia conversion into atmospheric N_2_. The potential removal of nitrates, ammonia, and sulfate is an important advantage of aerobic granules over AS [39].

While specific biological functions of aerobic granulation-compatible PFRs were inferred, the control samples involving activated sludge also exhibited distinct functionalities in their bacterial communities. When compared with PFR samples, AS samples showed significant enrichment in pathways involved in *purine metabolism* and *glycerophospholipid metabolism*. Both of which are responsible for the anabolism of amino acids and cell wall components [58,59]. Higher activity in these functional pathways is consistent with the promotion of bacterial growth in AS. This is further supported by the increased activity of *cysteine* and *methionine metabolisms* and their respective pathway enrichments. The *cysteine* and *methionine metabolic* pathways help improve the amino acid metabolism during digestion in waste activated sludge [60,61]. Furthermore, the higher prevalence of *aminoacyl-tRNA biosynthesis* and *ribosome biogenesis* function is indicative of higher protein synthesis, consistent with the faster bacterial metabolism in AS [62,63].

## 4. Conclusions, Limitations, and Future Prospects

Our metagenomic analysis revealed a clear distinction between the microbial community composition in each of the PFRs, which are characterized by different feast-to-famine ratios, and the AS biomass. As compared with the AS, granules produced in all three PFRs included a greater proportion of filamentous bacteria (e.g., *Sphaerolitus*) and bacteria producing large amounts of EPS (e.g., *Flavobacterium*), which are both essential for bacterial aggregation into stable granules. The filamentous genus *Sphaerolitus*, which was the most represented genus in the PFRs (18 to 33% of all bacteria), was more abundant in the 4- and 6-chamber PFRs than in the 8-chamber one, suggesting that too many filamentous cells may not be favorable to the development of dense, good-quality granules. The microbial diversity (based on Shannon index) was higher in the 8-chamber PFRs than in the 4- and 6-chamber reactors, which may be explained by the higher retention time in this reactor, leading to larger granules with distinct ecological niches at different depths from the granule surface. Functional analysis of the metagenomic data (PICRUSt2) revealed higher activity in secretion systems (e.g., *ABC transporters*) and signaling functions (e.g., *quorum sensing*) in the 8-chamber PFR, which is consistent with the cell functions observed in mature biofilms.

Although our study provided meaningful results, it is also associated with several limitations, which may be the starting point for future research. First, our analysis was based on DNA (metagenomics) and informed by the bacterial species and genera present in the system. Even though this allows some functional analysis based on the specific activities of bacterial groups, it is not a substitute for the analysis of the mRNA abundance (i.e., transcriptomics), which would provide a more detailed account of the microbial activities in the aerobic granule biomass. Another limitation of our study is that we collected the biomass only once during the operation of the PFRs. Even though the PFRs were operating in a steady state, our study does not account for the inevitable biomass fluctuation due to variation in the feeding wastewater. Moreover, our study does not elucidate the chronological changes in the microbial community during the development of the aerobic granules from the AS used to seed the PFRs.

The major contribution of this metagenomic study is to show that a leading factor in achieving aerobic granulation, i.e., the feast-to-famine ratio, induced clear changes in the microbial community structure, especially in the abundance of filamentous bacteria, which seemed to be adequately tuned to produce large and dense granules. We believe that our results will contribute to a deeper understanding of the mechanisms underlying the formation of aerobic granular biomass, which, despite its advantages, remains a challenge for real-world application in continuous wastewater treatment systems.

## Figures and Tables

**Figure 1 microorganisms-11-02328-f001:**
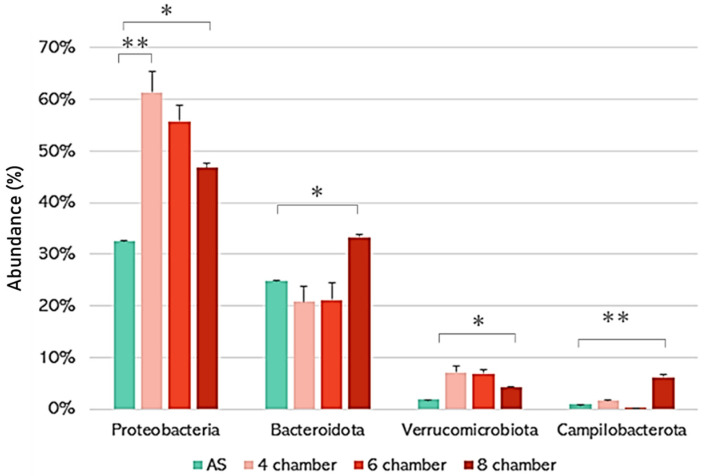
The proportion of the most prevalent phyla identified in the three PFRs and the AS control. Error bars represent the standard deviation between different chambers of each PFR. The stars indicate statistically significant differences in bacterial abundances: * *p* < 0.05; ** *p* < 0.01.

**Figure 2 microorganisms-11-02328-f002:**
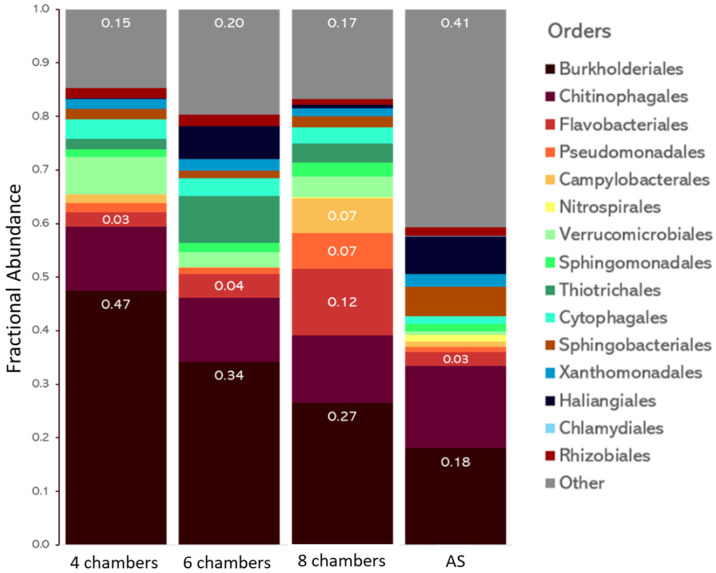
Fractional abundance (proportion of the total) of major bacterial orders observed in the three PFRs and the AS.

**Figure 3 microorganisms-11-02328-f003:**
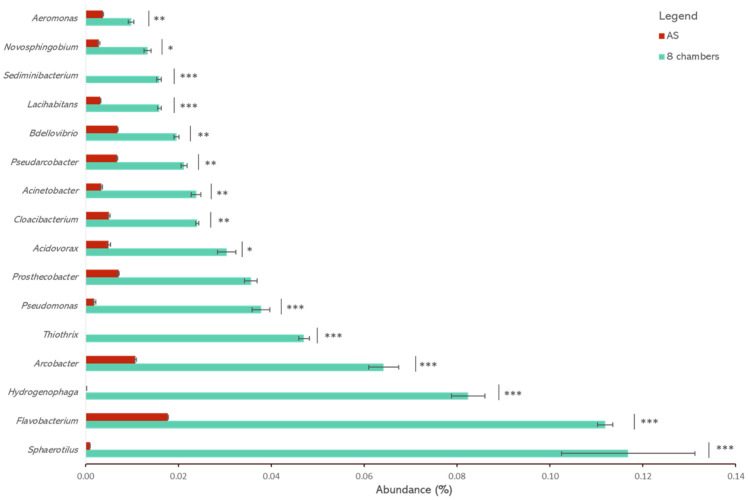
Microbial genera are significantly more abundant in aerobic granules in the 8-chamber PFR than in bacterial flocs in the AS control. Error bars represent the standard deviations between replicated samples. The stars indicate statistically significant differences between the 8-chamber PFR and AS: * *p* < 0.05, ** *p* < 0.01, *** *p* < 0.001.

**Figure 4 microorganisms-11-02328-f004:**
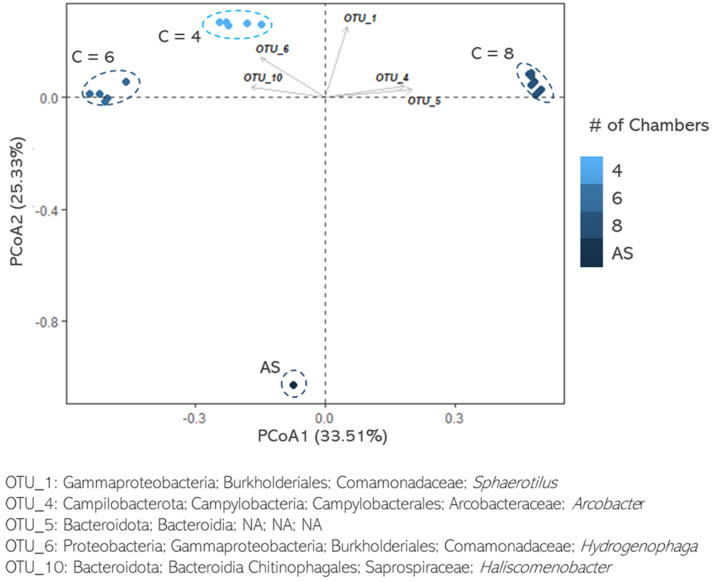
Principal coordinates analysis (PCoA) conducted on operational taxonomic units (OTUs) in each plug-flow reactor (PFR): 4 chambers (*n* = 6), 6 chambers (*n* = 8), and 8 chambers (*n* = 8), and the activated sludge control (AS). The three PFR systems and the AS control are clustered in dashed ellipses. The vectors of five main OTUs are shown.

**Figure 5 microorganisms-11-02328-f005:**
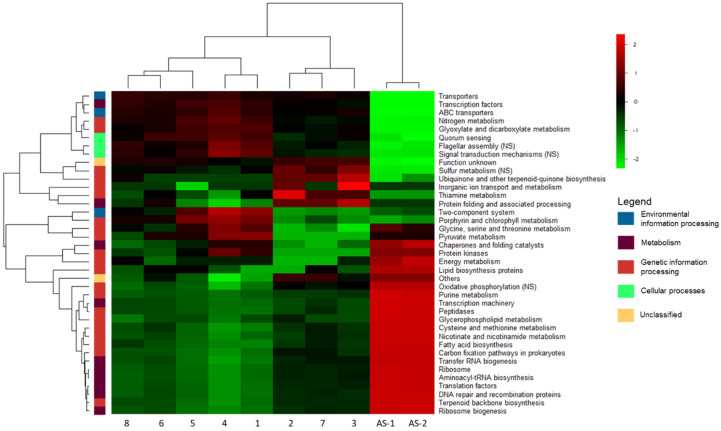
Heatmap of KEGG-predicted functions that are significantly different (*p* < 0.05) between the 8-chamber PFR and AS control samples. KEGG sub-pathways assigned to each function are color-coded.

**Table 1 microorganisms-11-02328-t001:** Diversity and richness indices of 16S rDNA sequences.

	Shannon Diversity Index (H)	Evenness (E)	Good’s Coverage	Simpson Index (D)	Chao 1	Number of Phylotypes or OTUs
4-chamber PFR (*n* = 6) ^a^	4.74 ± 0.09	0.78 ± 0.01	0.9961	0.972	456	436
6-chamber PFR (*n* = 8)	4.52 ± 0.27	0.77 ± 0.02	0.9970	0.965	377	362
8-chamber PFR (*n* = 8)	5.43 ± 0.20	0.80 ± 0.02	0.9996	0.985	906	896
Activated Sludge (*n* = 2)	6.28 ± 0.03	0.87 ± 0.00	0.9996	0.996	1324	1313
Nitrifying (*n* = 1)	3.45	0.60	1.0000	0.899	320	319

^a^ *n* = number of replicate samples.

## Data Availability

The data presented in this study may be available on request from the corresponding author. The data are not publicly available because the release of the data may require approval from the Upper Occoquan Service Authority, Centreville, VA, USA.

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
