# Peer review of "Metagenomic Analysis of a Continuous-Flow Aerobic Granulation System for Wastewater Treatment"

_microorganisms, 2023, doi:10.3390/microorganisms11092328_

Round 1

Reviewer 1 Report

Minor comments:

1- Lines 84-92 should be removed from the introduction section.

2 - Please, improve Figs 1 and 3's quality. 

3 - The remarks on this work should be included. Please, add a conclusion section and include future research scope.

Reviewer 2 Report

The text contains valuable and original results of metagenomic analyses of microorganisms used in continuous-flow aerobic granulation water treatment.

The research is well conducted and comprehensively described.

However, the following comments occurred to the reviewer while reading:

- The graph in Figure 2 should describe the variable from the OY axis.

- The article lacks a separate chapter with a summary and clearly stated conclusions.

Once these deficiencies are corrected, the text can be published and will be a valuable source of information for researchers involved in using aerobic granulation in wastewater treatment.

Reviewer 3 Report

The manuscript concerns the important issue of the metagenomic analysis of a continuous-flow aerobic granulation system for wastewater treatment. In the research, authors conducted a metagenomic investigation with the aim of describing the bacterial makeup of granular biomass formed in three simulated plug flow reactors (PFRs). These reactors had varying feast-to-famine ratios. The phylogenetic examinations showed a clear differentiation in bacterial composition between aerobic granules in the experimental PFRs and the traditional activated sludge method. Authors noticed that in the PFR with the longest period of famine, there were larger, more compact granules that settled better. These granules contained a higher proportion of bacteria that produce abundant extracellular polymeric substances (EPS). By using functional metagenomic analysis based on KEGG pathways, authors found that the larger, denser aerobic granules in the PFR with the longest famine period exhibited increased functionalities linked to secretion systems and quorum sensing. These traits are commonly associated with bacteria found in biofilms and aerobic granules. This study adds to comprehension of how aerobic granule morphology relates to the bacterial composition within the granular biomass. The paper provides substantive discussion. The study is applicable and provide insight into accuracy performance.  Please provide comparisons with other models and limitations of the presented study. Why you chose these particular approach for your research? Also can you explain the advantages of the presented approach? What is the significance of your contribution?

Reviewer 4 Report

The manuscript presented by the authors shows a metagenomic analysis of a continuous flow aerobic granulation system for wastewater treatment; with this research, the authors seek to contribute to the knowledge of the relationship between the morphology of the aerobic granules and the bacterial composition of the granular biomass that allows obtaining a better effluent at the end. It is a relevant topic to solve the problems of many wastewater treatment plants using aerobic processes, mainly activated sludge with continuous complete mixing systems, where there is a considerable generation of sludge, which in some cases is complex to settle and remove from the treated effluent. However, there are some considerations to be taken into account, which are outlined below:

1. In the introduction, the authors could give a context of the different methods currently used to solve the problem of secondary sedimentation of activated sludge or aerobic processes where there is considerable biomass generation. It is not evident from the introduction what technologies exist, what problems exist, and how this research addresses them.

2. It is striking that the authors contextualize the problem in continuous flow reactors (CFR) and the study is done with piston flow reactors; it is suggested to improve the explanation of this aspect, as clearly, these two types of process have very different hydrodynamic characteristics, mass transfer and cell retention times, as well as different growth models. It is not clear why they did not perform the research on CFR reactors.

3. It is suggested that the authors expand and clearly state the contribution of this work to knowledge and that it allows not only to understand the process that contributes to aerobic granulation but also how strategies can be generated to improve secondary sedimentation in aerobic reactors.

4. In the section Simulated PFR characteristics and local domestic wastewater, it is suggested that the authors make a diagram or figure that allows a clear understanding of the process. It is recommended to include the amount of dissolved oxygen in the system, the temperature that was worked, and the flow rate of the process.

5. It is suggested to include the times when the samples were taken in the methodology. This is important because the microbial diversity of the sludge changes over time, which is associated with the type and quantity of nutrients present in the waste effluent.

6. It is suggested that the authors show a characterization of the wastewater used, and it is also essential to show the physicochemical characterization at the time of sampling, given that the presence or absence of each type of microorganism is associated with the amount of nutrients.

7. The results are theoretically supported, and the genera found are associated with existing reports on the production of exopolysaccharides. However, there is no evidence of this measurement in this study. It is suggested that the authors include these results, if available, to correlate them with the presence of families in each of the samples analyzed.

8. Improve the resolution of figures 2, 3, 4, and 5.

9. The results should be further discussed and contrasted with existing literature.

10. No conclusions are presented 

Round 2

Reviewer 3 Report

Accept in the present form.

Reviewer 4 Report

A revised version of the manuscript submitted by the authors shows that the comments made were taken into account and substantially improved the document. It is again recommended that the process diagram be added to the manuscript. Finally, it is recommended that the process for publication of the manuscript be continued.